# Proteomic Assessment of Hypoxia-Pre-Conditioned Human Bone Marrow Mesenchymal Stem Cell-Derived Extracellular Vesicles Demonstrates Promise in the Treatment of Cardiovascular Disease

**DOI:** 10.3390/ijms24021674

**Published:** 2023-01-14

**Authors:** Cynthia M. Xu, Catherine Karbasiafshar, Rayane Brinck Teixeira, Nagib Ahsan, Giana Blume Corssac, Frank W. Sellke, M. Ruhul Abid

**Affiliations:** 1Cardiovascular Research Center, Rhode Island Hospital, Providence, RI 02903, USA; 2Division of Cardiothoracic Surgery, Alpert Medical School of Brown University, Rhode Island Hospital, Providence, RI 02903, USA; 3Mass Spectrometry, Proteomics and Metabolomics Core Facility, Stephenson Life Sciences Research Center, University of Oklahoma, Norman, OK 73019, USA; 4Department of Chemistry and Biochemistry, University of Oklahoma, Norman, OK 73019, USA; 5Cardiovascular Physiology Laboratory, Basic Health Sciences Institute, UFRGS, Porto Alegre, RS, Brazil

**Keywords:** human bone marrow mesenchymal stem cell-derived extracellular vesicles, starvation, hypoxia, pre-conditioning, cardiovascular disease

## Abstract

Human bone marrow mesenchymal stem cell derived-extracellular vesicles (HBMSC-EV) are known for their regenerative and anti-inflammatory effects in animal models of myocardial ischemia. However, it is not known whether the efficacy of the EVs can be modulated by pre-conditioning of HBMSC by exposing them to either starvation or hypoxia prior to EV collection. HBMSC-EVs were isolated following normoxia starvation (NS), normoxia non-starvation (NNS), hypoxia starvation (HS), or hypoxia non-starvation (HNS) pre-conditioning. The HBMSC-EVs were characterized by nanoparticle tracking analysis, electron microscopy, Western blot, and proteomic analysis. Comparative proteomic profiling revealed that starvation pre-conditioning led to a smaller variety of proteins expressed, with the associated lesser effect of normoxia versus hypoxia pre-conditioning. In the absence of starvation, normoxia and hypoxia pre-conditioning led to disparate HBMSC-EV proteomic profiles. HNS HBMSC-EV was found to have the greatest variety of proteins overall, with 74 unique proteins, the greatest number of redox proteins, and pathway analysis suggestive of improved angiogenic properties. Future HBMSC-EV studies in the treatment of cardiovascular disease may achieve the most therapeutic benefits from hypoxia non-starved pre-conditioned HBMSC. This study was limited by the lack of functional and animal models of cardiovascular disease and transcriptomic studies.

## 1. Introduction

Throughout the last decade, stem cell therapies have been extensively studied in human clinical trials to treat cardiac ischemia, myocardial infarction, and heart failure. Such trials include the 2013 C-CURE (Cardiopoietic stem Cell therapy in heart failure) multicenter randomized trial, the 2013 CADUCUES (CArdiosphere-Derived aUtologous stem CElls to reverse ventricUlar dysfunction) trial, the 2017 CHART-1 (Congestive Heart Failure Cardiopoietic Regenerative Therapy) trial, the 2017 BOOST-2 (BOne marrOw transfer) trial, and the 2022 Prevent-TAHA8 (Transplantation of mesenchymal stem cells for prevention of acute myocardial infarction induced heart failure) phase III trial [1,2,3,4,5]. Thus far, these studies have shown the safety of stem cell therapy. Further data on clinical superiority will be revealed with the Prevent-TAHA8 phase III trial, which will continue its data collection until November 2024. However, in terms of therapeutic efficacy, clinical improvements do not appear to be sustained long-term [3,6]. This may be due to the low cell migration, tissue retention, and survival rate in the ischemic myocardium [7]. There is also a risk of tumorigenicity from the use of mesenchymal stem cells, especially for cancer therapy, but this has not been reported in cardiac disease [8,9].

However, extracellular vesicles secreted by the stem cells may overcome some of these challenges, as these vesicles can cross barriers or membranes very easily, including the blood-brain barrier, and have no tumorigenic potential [10]. Human bone marrow mesenchymal stem cell-derived extracellular vesicles (HBMSC-EV) are lipid-bilayer vesicles released from human bone mesenchymal stem cells (HBMSC) and thought to be responsible for cell communication and the beneficial paracrine effects of the stem cells [11,12]. HBMSC-EV contain a mixture of proteins, RNA, DNA, and lipids, which have been largely demonstrated to have regenerative and anti-inflammatory properties with improvements in cardiac function in both small and large animal models of cardiac ischemia [11,13,14,15,16,17,18,19]. Additionally, there is an ongoing clinical trial evaluating the safety of extracellular vesicles after acute myocardial infarction [20]. One of the main barriers to extracellular vesicle use for cardiac disease is effective non-invasive delivery to the heart—currently, cardiac homing techniques are being investigated [21,22,23,24].

HBMSC-EV properties can be modified through pre-conditioning of the HBMSC with either starvation or hypoxia pre-conditioning, yet it is unclear at this time which conditions are more beneficial for cardiovascular disease (CVD). The vast majority of studies have used either hypoxia or starvation pre-conditioned HBMSC-EVs, but not in direct comparison with each other [15,19,25,26,27]. Moreover, HBMSC-EV isolation protocols and research are limited by heterogeneity. HBMSC-EV properties are dependent on the protocols by which they were isolated and given the many components that HBMSC-EVs contain, omics-type studies have an important role in characterizing HBMSC-EV properties and function. Proteomics has been used to study HBMSC-EVs and have identified proteins involved in pathways of cellular proliferation, adhesion, migration, morphogenesis, and angiogenesis after hypoxia pre-conditioning. However, the effects of different types of EV pre-conditioning on CVD, including myocardial ischemia, is not known [28,29,30,31]. Thus, the main objective of this study was to characterize the effects of hypoxia and starvation pre-conditioning on the protein cargo of HBMSC-EVs to determine their potential role in CVD.

## 2. Materials and Methods

### 2.1. HBMSC Culture

HBMSC were purchased from Lonza (Walkersville, MD, USA, PT-2501), and were cultured in T150 cm^2^ flasks with Dulbecco’s Modified Eagle Medium (DMEM) (Gibco #11965092), 10% fetal bovine serum (Gibco #16140063) and 1% penicillin-streptomycin (Gibco #15140-122) to passage 7. The cells were grown in a humidified incubator at 37 °C with 5% CO_2_. At passage 7, the cells designated for normoxia pre-conditioning were plated onto T150 cm^2^ flasks (media volume 18 mL), and the cells designated for hypoxia pre-conditioning were plated onto 10 cm dishes (media volume 10 mL).

### 2.2. Normoxia Starvation (NS) HBMSC-EV Isolation

Once the HBMSC (T150 cm^2^ flasks) reached 80% confluence at passage 7, the regular growth media was removed (18 mL), the cells were washed with 15 mL of Dulbecco’s Phosphate Buffered Saline (PBS) (Gibco #14190144), and 18 mL Roswell Park Memorial Institute (RPMI) (Gibco #11875085) was added. The cells were placed in the incubator (37 °C with 5% CO_2_), and 24 h later, the media was collected. The media was then centrifuged at 2000× *g* twice to remove the cell debris. The media was then centrifuged at 100,000× *g* (WX Ultra Centrifuge with Sorvall AH-629 rotor) for 70 min to isolate the HBMSC-EVs, then washed with PBS twice with additional 70 min centrifuge cycles at 100,000× *g*. The NS HBMSC-EVs were re-suspended in PBS with 1% dimethylsulfoxide (DMSO) and stored at −80 °C.

### 2.3. Normoxia Non-Starvation (NNS) HBMSC-EV Isolation

Once the HBMSC (T150 cm^2^ flasks) reached 80% confluence at passage 7, the regular growth media was removed and replaced with 18 mL DMEM. The cells were placed in the incubator (37 °C with 5% CO_2_), and 24 h later, the media was collected. The media was then centrifuged at 2000× *g* twice to remove the cell debris. The media was then centrifuged at 100,000× *g* for 70 min to isolate the HBMSC-EVs, then washed with PBS twice with additional 70 min centrifuge cycles at 100,000× *g*. The NNS HBMSC-EVs were re-suspended in PBS with 1% DMSO and stored at −80 °C.

### 2.4. Hypoxia Starvation (HS) HBMSC-EV Isolation

Once the HBMSC (10 cm dish) reached 80% confluence at passage 7, the regular growth media was removed (10 mL), the cells were washed with 10 mL of PBS, and 10 mL of RPMI was added. The cells were then placed in an airtight humidified hypoxia chamber (Billups-Rothenberg, MIC-101) containing 95% N_2_ and 5% CO_2_. Hypoxia was induced by connecting the chamber’s inflow cannula to a gas tank containing 95% N_2_ and 5% CO_2_ with a flow rate of 20 L/minute with the outflow cannula open for 7 min to wash out O_2_. After 7 min, the outflow cannula was clamped shut first, then the inflow cannula was clamped, and the gas flow was turned off. The chamber was then placed at 37 °C for 24 h. Afterwards, the hypoxia chambers were opened, and the media was collected and centrifuged at 2000× *g* twice to remove the cell debris. The media was then centrifuged at 100,000× *g* for 70 min to isolate the HBMSC-EVs, then washed with PBS twice with additional 70 min centrifuge cycles at 100,000× *g*. The HS HBMSC-EVs were re-suspended in PBS with 1% DMSO and stored at −80 °C.

### 2.5. Hypoxia Non-Starvation (HNS) HBMSC-EV Isolation

Once the HBMSC (10 cm dish) reached 80% confluence at passage 7, the regular growth media was removed (10 mL), and fresh 10 mL DMEM was added. Hypoxia was induced within the hypoxia chambers, and the cells were incubated at 37 °C for 24 h. Afterwards, the hypoxia chambers were opened, and the media was collected and centrifuged at 2000× *g* twice to remove the cell debris. The media was then centrifuged at 100,000× *g* for 70 min to isolate the HBMSC-EVs, then washed twice with PBS with additional 70 min centrifuge cycles at 100,000× *g*. The HNS HBMSC-EVs were re-suspended in PBS with 1% DMSO and stored at −80 °C.

### 2.6. HBMSC-EV Characterization

The size, concentration, and distribution of HBMSC-EVs was determined by nanoparticle tracking analysis (NTA) using the NanoSight NS500 (Malvern Instruments, Malvern, UK). The NanoSight was first verified and optimized with control beads (Malvern Instruments). The HMBSC-EV sample dilutions were prepared in accordance with the manufacturer’s instructions, and samples were analyzed in triplicates. To prepare the HBMSC-EVs for electron microscopy (FEI Morgagni 268), the HBMSC-EVs were fixed in 2% paraformaldehyde and placed onto formvar-coated electron microscope grids. After 20 min, the HBMSC-EVs were washed with PBS, fixed with 1% glutaraldehyde, and contrasted in 4% uranyl acetate for imaging. The following HBMSC-EV markers were identified on Western blot analysis: CD81 (Cell Signaling, Danvers, MA, USA, #52892S), CD9 (Cell Signaling, #13403S), Alix (Cell Signaling, #92880S), GAPDH (Cell Signaling, #97166S), heat shock protein 70 (Cell Signaling, #4872T), and albumin (Cell Signaling, #4929S) [32]. The HBMSC-EV were lysed using RIPA Buffer (Sigma Aldrich, Burlington, MA, USA, #R0278-50ML) and Protease Inhibitor Cocktail (Sigma Aldrich, #P8340-1ML). The protein concentrations were quantified using the Micro BCA Protein Assay Kit (Thermo Scientific, Waltham, MA, USA, #23235). Then, 10 µg of HBMSC-EV protein were reduced and denatured at 70 °C for 10 min and then immediately loaded onto the gel.

### 2.7. Proteomic Analysis of HBMSC-EV

The NS, NNS, HS, and HNS HBMSC-EVs samples were prepared using three independent isolations of HBMSC-EV, and analyses were performed in triplicates (n = 3). The LC-MS/MS analysis was performed as described previously [33]. Peptide spectrum matching of MS/MS spectra of each file was searched against UniProt protein database using the Sequest algorithm within Proteome Discoverer (PD) v 2.3 software (Thermo Fisher Scientific, San Jose, CA, USA). The relative label-free quantitative and comparative analysis of the samples was performed using the Minora algorithm and the adjoining bioinformatics tools of the Proteome PD 2.3 software. Various open-source bioinformatic platforms, such as ShinyGO 0.76 (http://bioinformatics.sdstate.edu/go/ (accessed on 30 December 2022)), Venney (https://bioinfogp.cnb.csic.es/tools/venny (accessed on 30 December 2022)), and Morpheus (https://software.broadinstitute.org/morpheus (accessed on 30 December 2022)), were used for analyzing the proteomics dataset.

### 2.8. Western Blot Analysis of HBMSC

Lysates of the HBMSC after no pre-conditioning, starvation pre-conditioning, and hypoxia pre-conditioning were submitted for Western blot analysis. The following antibodies were used: Hypoxia-inducible fact 1α (HIF1-α) (Cell Signaling, #36169), phospho-AMP-activated protein kinase (Thr172) (p-AMPK) (Cell Signaling, #50081), AMP-activated protein kinase (AMPK) (Cell Signaling, #5831), phospho-Akt (Thr308) (p-Akt) (Cell Signaling, #13038), Akt (Cell Signaling, #4685), phospho-mTOR (Ser2481) (p-mTOR) (Cell Signaling, #2974), mTOR (Cell Signaling, #2983), phospho-p44/42-mitogen-activated protein kinase (Thr202/Tyr204) (p-ERK) (Cell Signaling, #4377), p44/42 mitogen-activated protein kinase (ERK) (Cell Signaling, #4695), and GAPDH. Statistical analyses were performed with the Kruskal–Wallis and post-hoc Dunn’s statistical tests.

## 3. Results

### 3.1. HBMSC-EV Characterization

Via NTA, the NS, NNS, HS, and HNS HMBSC-EV mean particle sizes were determined to be 228.2 ± 31.6, 194.9 ± 11.9, 415.7 ± 20.3, and 179.2 ± 7.3 nm, respectively (Figure 1A). NS, NNS, and HNS HBMSC-EV particles were characterized by electron microscopy (Figure 1B). Western blot analyses demonstrated the presence of the transmembrane proteins CD81 and CD91, and cytosolic proteins Alix and GAPDH in NS, NNS, and HNS HMBSC-EVs. HSP70, a promiscuous cytosolic protein, was only identified in the HNS EV subtype. Albumin was not identified in any of the HMBSC-EVs, confirming the purity of EVs harvested (Figure 1C). We were not able to characterize HS HBMSC-EV with electron microscopy or Western blot due to very low levels of EVs produced when HBMSC were exposed to both nutrition (starvation) and oxygen-depleted (hypoxia) conditions.

### 3.2. Hypoxia and Normoxia Have Disparate Effects without Starvation Pre-Conditioning

Principal component analysis (PCA) of the HBMSC-EV proteins showed that when the cells underwent starvation pre-conditioning, normoxia, or hypoxia pre-conditioning did not change the protein contents of the cells significantly. However, when HBMSC was incubated in rich media, normoxia (NNS) and hypoxia (HNS) HBMSC exposure resulted in NNS and HNS HBMSC-EV contents that were different from each other (Figure 2A). Heat map analyses demonstrated a distinct protein signature for each HBMSC-EV subtype. Within the common proteins identified, the HNS subtype was the most enriched in specific groups of proteins (Figure 2B).

Overall, HNS HBMSC-EV had the greatest protein variety, with a total of 395 proteins identified with 74 unique proteins. NNS HBMSC-EV had the second most total proteins, with 367 proteins identified and 14 unique proteins. The NS and HS HBMSC-EV subtypes had the lowest number, with 276 and 235 total proteins, respectively (Figure 3A). A total of 172 proteins were common in all HBMSC-EV types (Figure 3A). Pathway analysis of these proteins showed pathways with regenerative or adaptive roles, such as “carbon metabolism,” “biosynthesis of amino acids,” “HIF-1 signaling pathway,” and “ribosome”, were enriched (Figure 3B). Furthermore, pathway analysis of the 74 unique HNS proteins demonstrated that pathways involved in angiogenesis, such as “cell migration,” “cell adhesion,” and numerous inflammatory pathways, were enriched (Figure 3C).

Proteins involved in redox, calcium handling, transport, and metabolism pathways were identified, and heat map analysis showed that NNS and HNS HBMSC-EV subtypes were comparatively enriched in these proteins (Figure 3D). For instance, HNS HBMSC-EV had the greatest overall quantity and variety of redox and metabolism proteins. HNS and NNS subtypes had a similar amount of calcium-handling proteins identified. Solute carrier family 2 member 1 (SLC2A1), or glucose transporter protein type 1 (GLUT1), was identified in all HBMSC-EV subtypes. Many ribosomal proteins were identified as well (not shown)—the NNS HBMSC-EV subtype had the greatest quantity and variety of these.

### 3.3. Hypoxia Pre-Conditioning Has Significant Effects on Stress Response and Proliferative Signaling Pathways

Western blots showed that hypoxia pre-conditioning induced several changes in major stress response and growth signaling pathways of the HBMSC (Figure 4). Hypoxia pre-conditioning resulted in increased AMPK activation and decreased Akt activation. HIF1-α expression was significantly increased after hypoxia pre-conditioning.

## 4. Discussion

Among the four different HBMSC-EVs isolated in this study, hypoxia pre-conditioning of HBMSC without starvation resulted in the most significant variety of HBMSCC-EV proteins overall with the highest number of redox proteins and a proteomic pathway profile suggestive of improved angiogenesis. Interestingly, under starvation pre-conditioning, normoxia or hypoxia did not appear to influence the HBMSC-EV proteomic profile as greatly as that of non-starved HBMSC.

Proteomics identified the greatest cargo of redox and metabolism-related proteins in HBMSC-EV after hypoxia pre-conditioning, as well as proteins involved in calcium handling. A number of redox enzymes such as catalase, superoxide dismutase 1, and several peroxiredoxins have known roles in lowering ROS. All of the HBMSC-EV subtypes contained GLUT1, the most abundant glucose transporter in the heart and is important for basal cardiac glucose uptake [34,35]. The consequence of the delivery of metabolic enzymes such as glycolytic enzymes or calcium handling proteins such as sorcin to the myocardium is not clear. However, HBMSC-EV and their contents represent a milieu of -cell communication and adaptive response of the HBMSC to cellular stress or pre-conditioning and could confer advantages to their target tissues. In the absence of oxygen, such as in the ischemic myocardium, delivery of glycolytic enzymes such as pyruvate kinase and lactate dehydrogenase or glucose transporters may affect glucose metabolism.

Thus far, studies of animal models mimicking ischemic cardiac disease have used hypoxia-pre-conditioned mesenchymal stem cells with good results [36]. Hypoxia-pre-conditioned extracellular vesicles have been found to be enriched in multiple pro-angiogenic miRNA, such as miR-125b-5p, miR-21-5p, miR-21, miR-1246, miR-23a-3p, and miR-23 [19,37,38]. Another study demonstrated improved survival, infarct size, and cardiac function after intramyocardial injection of hypoxia-pre-conditioned extracellular vesicles compared to that of normoxia pre-conditioned extracellular vesicles with corresponding changes in the miRNA profile [39]. There has also been an investigation into HIF1-α-over-expressing stem cells and their resulting extracellular vesicles, which showed improved neovessel formation and heart function in a rat myocardial infarction model [40].

In conclusion, this study demonstrated that hypoxia pre-conditioning of HBMSC without starvation induces the most significant variety of HBMSC-EV proteins with the highest number of redox proteins and a proteomic pathway profile suggestive of improved angiogenesis. These findings also suggest that future HBMSC-EV studies on the treatment of cardiovascular disease may achieve the most therapeutic benefits from hypoxia-pre-conditioned HBMSC. With the potential of improved angiogenesis from hypoxia-derived HBMSC-EV, these HMBSC-EV may be best suited for cardiovascular conditions with diminished blood flow or capillary density, such as myocardial ischemia or heart failure.

## 5. Study Limitations

One limitation of this study was a lack of functional and animal models of cardiovascular disease to compare NS, NNS, HNS, and HS HBMSC-EV side by side. Ongoing studies in our lab are addressing this issue using small and large animal models of myocardial ischemia and HBMSC-EV. Additionally, transcriptomic studies should be performed in the future as RNA plays a major role in HBMSC-EV function.

## Figures and Tables

**Figure 1 ijms-24-01674-f001:**
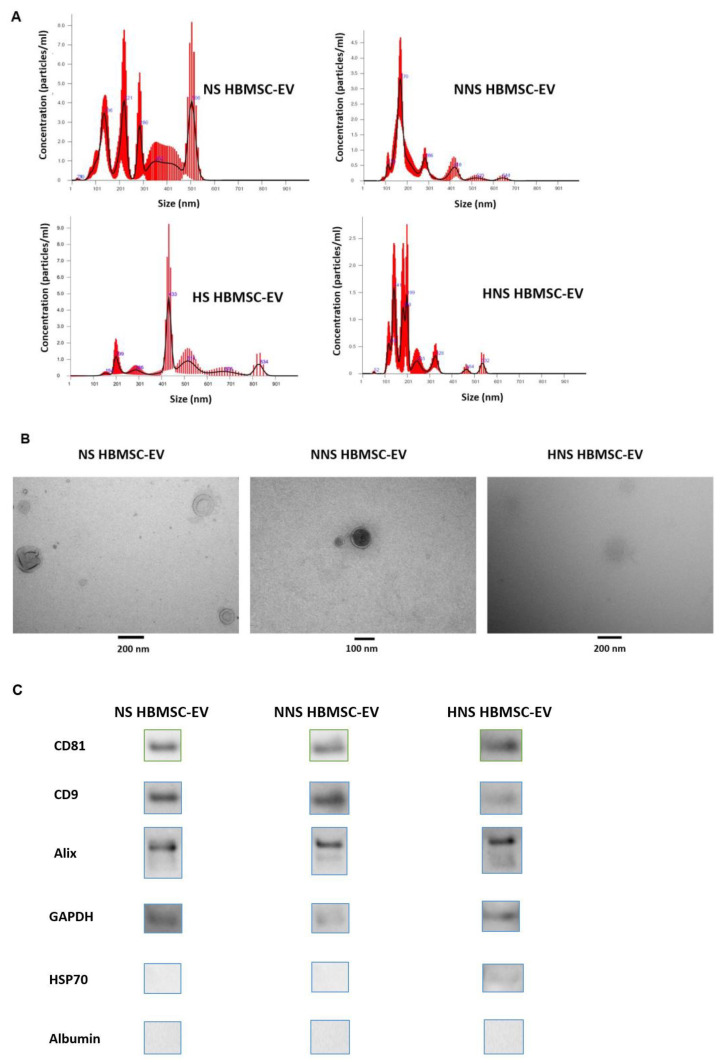
NS, NNS, HS, and HNS HBMSC-EV Characterization. Characterization was done with (**A**) NTA, (**B**) electron microscopy, and (**C**) Western blot. CD81, CD9, Alix, and GAPDH were identified in the NS, NNS, and HNS HBMSC-EV types. HSP70 was only identified in HNS HBMSC-EV. Albumin was not identified in any of the HBMSC-EV types, the absence of which is a marker of sample purity. Western blot was used to only qualitatively identify the presence of known HBMSC-EV markers and no statistical analysis or comparisons were performed between HBMSC-EV types. Western blot bands are displayed separately, given the need for wide-ranging exposure times. Given the extremely low protein content of HS HBMSC-EV, electron microscopy and Western blot characterization were not able to detect this HBMSC-EV. Each experiment was performed in triplicates using three independently isolated EV samples (n = 3).

**Figure 2 ijms-24-01674-f002:**
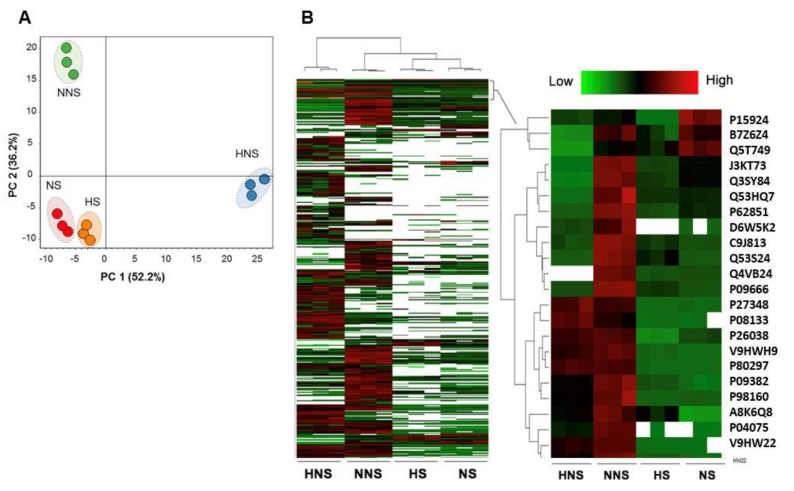
Proteomic analysis demonstrated qualitative and quantitative differences in HBMSC-EV protein content when the HBMSC were not exposed to starvation pre-conditioning. (**A**) Principal component analysis (PCA) of normalized total protein abundance (peak area) of the HBMSC-EV subtypes showed that the protein signatures of the NS (normoxia starvation) and HS (hypoxia starvation) subtypes were similar, while the rich media HBMSC-EV subtypes exposed to normoxia (NNS) and hypoxia (HNS) were significantly different from each other. (**B**) Heat map clustering of the total proteins identified and quantified proteins shows differential abundance among the four HMBSC-EV subtypes that had a distinct enrichment pattern.

**Figure 3 ijms-24-01674-f003:**
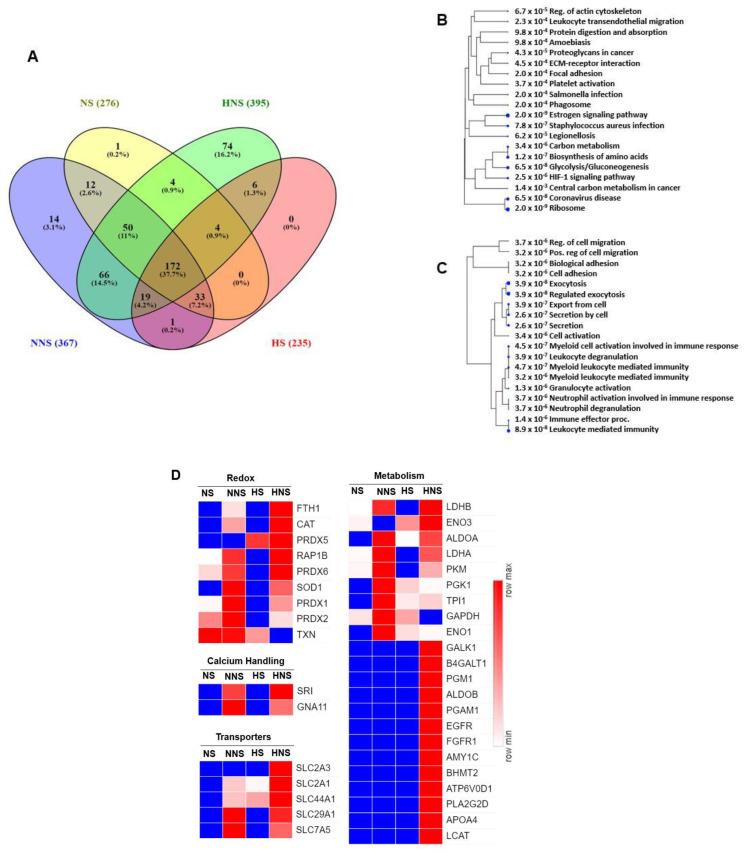
Qualitative and quantitative analysis of HBMSC-EV proteins identified. (**A**) Venn diagram analysis shows the unique and overlapped proteins identified in the NS, NNS, HS, and HNS HBMSC-EVs samples. (**B**) Hierarchal tree pathway analysis of common proteins (172) found in all HBMSC-EV subtypes, which identified pathways related to regeneration and adaptation, such as “carbon metabolism”, “biosynthesis of amino acids”, “HIF-1 signaling pathway”, and “ribosome”. (**C**) Hierarchal tree pathway analysis of 74 proteins unique to HNS HBMSC-EVs, which identified pathways that may be beneficial to angiogenesis. (**D**) Heat map shows the total protein abundance of targeted protein sets. Categories of proteins identified, which demonstrated that HNS HBMSC-EVs were the most enriched in redox and metabolism-associated proteins. Both NNS and HNS HBMSC-EV contained proteins associated with calcium handling. SLC2A1, or GLUT1, was identified in all HBMSC-EV types. Numbers represent arbitrary units and relative quantity.

**Figure 4 ijms-24-01674-f004:**
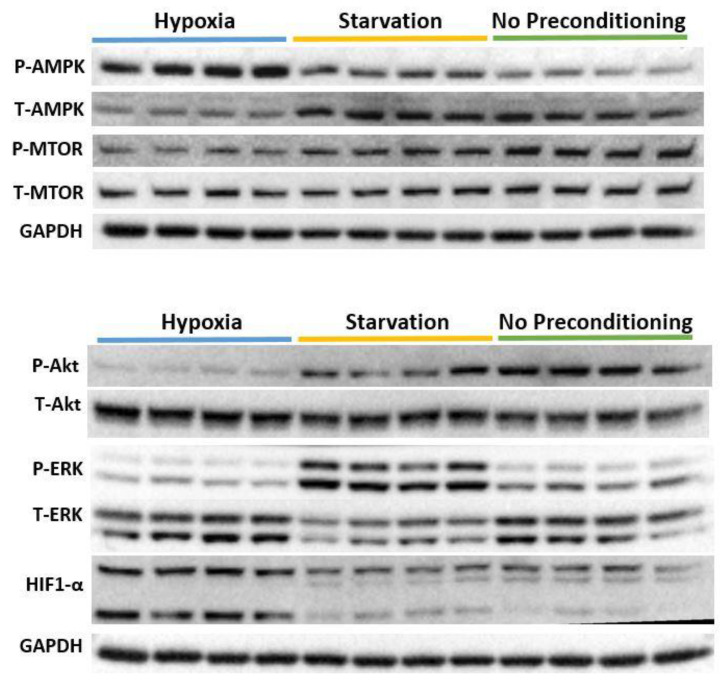
Western blot analyses of HBMSC after either hypoxia (n = 4) or starvation (n = 4) pre-conditioning showed that hypoxia and starvation pre-conditioning significantly modulated major stress response and proliferative pathways compared to cells under normoxic non-starved (n = 4) conditions. HIF1-alpha expression was significantly increased after hypoxia pre-conditioning.

## Data Availability

The data that support the findings of this study are available upon request from the first author, C.M.X.

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
