# Peer review of "Proteomic Assessment of Hypoxia-Pre-Conditioned Human Bone Marrow Mesenchymal Stem Cell-Derived Extracellular Vesicles Demonstrates Promise in the Treatment of Cardiovascular Disease"

_ijms, 2023, doi:10.3390/ijms24021674_

Round 1

Reviewer 1 Report

This is a very interesting article opening new avenues to help people with cardiac ischemia. Since the subject is rather new, I would like more general facts in the introduction.

Line 160: you say that there are 4 preconditioning situations, but why NNS ?! non starvation and non hypoxia is a not preconditioned line, isn`t it?! Before , in abstract, you describe pre-conditioning as exposure to hypoxia and/or starvation. Please, decide. 

Reviewer 2 Report

The authors have conducted a novel and interesting research about the potential of Hypoxia Pre-conditioned Human Bone Marrow Mesenchymal Stem Cell-Derived Extracellular Vesicles Cardiovascular Diseases treatment.

Major recommendation:

In order to have continuity and better understanding I suggest to describe first 2. Materials and Methods, then Results and Discussion.

Also, I suggest to separate your Conclusions and Study limitation in different sections, for a better overview of your research.

In what cardiovascular diseases the authors believe that HBMSC-EV would be mostly effective? If possible, I suggest to introduce in your discussion this point of view.

Minor recommendation:

Attention to all abbreviations and their meanings as they should be inserted as they first appear in text; for examples missing explanation for AMPK, GLUT1, HIF1 etc, check the manuscript.

All the paragraph should be with justify.

Round 2

Reviewer 2 Report

Dear Authors,

I appreciate your interest in the suggestions made, and I hope I've helped improve your article.